# A universal compression theory for lottery ticket hypothesis and neural scaling laws

**Hong-Yi Wang**
Princeton University & NTT Research
hywang@princeton.edu

**Di Luo**
Tsinghua University & UCLA
diluo@tsinghua.edu.cn

**Isaac L. Chuang**
MIT
ichuang@mit.edu

**Tomaso Poggio**
MIT
tp@ai.mit.edu

**Liu Ziyin**
MIT & NTT Research
ziyinl@mit.edu

## Abstract

When training large-scale models, the performance typically scales with parameter count and dataset size via a slow power law. We show that comparable performance can be achieved with far smaller models and much less data. We prove that a generic permutation-invariant function of $d$ objects can be compressed into a function of $\mathrm{polylog}\, d$ objects with vanishing error, and that this rate is optimal. This yields two key implications: (Ia) a large neural network can be compressed to polylogarithmic width while preserving its learning dynamics; (Ib) a large dataset can be compressed to polylogarithmic size while leaving the model's loss landscape unchanged. (Ia) establishes the *dynamical* lottery ticket hypothesis: ordinary networks can be strongly compressed without changing learning dynamics or outcomes. (Ib) shows that a scaling law $L \sim d^{-\alpha}$ can be accelerated to arbitrarily fast power-law decay, and ultimately to $\exp\left(-\alpha' \sqrt[m]{d}\right)$.

## 1 Introduction and Problem Setup

Training modern neural networks requires enormous computational and data resources. A central empirical observation is that performance improves with scale according to slow neural scaling laws, where the error decays as a power of dataset size or model width,

$$L(N) \propto N^{-\alpha}, \tag{1}$$

with small exponents $\alpha \in [0.1, 0.3]$ for large language models (Kaplan et al., 2020). Such slow decay implies that achieving even modest performance gains requires orders-of-magnitude increases in data or parameters, raising the question of whether current learning systems are fundamentally inefficient, or whether their effective complexity is much lower than their apparent size.

In this work, we provide a geometric explanation for this inefficiency and a constructive resolution. We show that a broad class of functions relevant to modern machine learning—including losses, predictions, and training dynamics—are *permutation invariant* in large collections of objects, e.g., data points, neurons, attention heads. Say there are $d$ objects, each lying in a linear space $V = \mathbb{R}^m$. Viewed geometrically, these functions are functions not on $V^d$, but on the quotient space $V^d/S_d$. Exploring the symmetry and the resulting quotient space leads to drastic improvement to model training efficiency. Specifically, such functions can be approximated by only $O(\mathrm{polylog}(d))$ objects.

This universal compression result has two main consequences. First, it implies a *dynamical lottery ticket hypothesis*: wide neural networks can be compressed to polylogarithmic width while preserving their entire training dynamics. Second, it provides a mechanism for improving neural scaling laws by replacing large datasets or models with much smaller, geometrically equivalent representatives. Together, these results show that large models often operate in a regime of extreme redundancy induced by symmetry, and that respecting this geometric structure enables substantially simpler representations.

A rigorous formulation and relevant proofs are deferred to the Appendix.

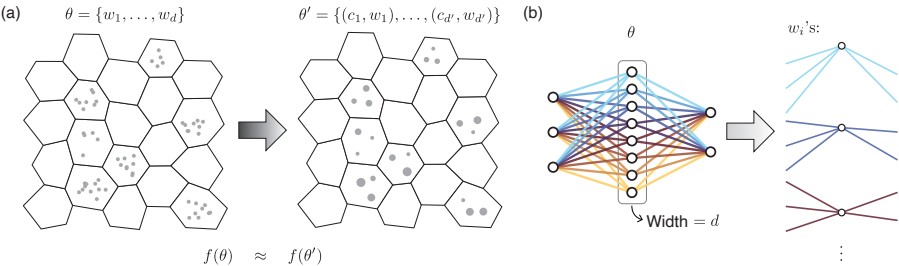

Figure 1: (a) Illustration of the main idea behind the compressibility of neural networks and datasets. (1) Permutation symmetry allows a high-dimensional function to be decomposed into a composition of $d$ low-dimensional "objects" (dots in the figure). (2) When $d$ is large, these objects become crowded, and those lying in denser regions are essentially redundant; they can be compressed into $d' = O(\mathrm{polylog}\, d)$ objects. The potential curse of dimensionality can thus be mitigated, or even removed, when the underlying function is smooth—a lesson well known in nonparametric statistics. (b) Decomposing the linear weights of a neural network into "objects" of symmetric status.

We start with the problem setup. A function $f : V^d \to \mathbb{R}$ is *permutation invariant* (or simply *symmetric*) if $f(\dots, w_i, \dots, w_j, \dots) = f(\dots, w_j, \dots, w_i, \dots)$ for any $i \neq j$. The theory in this paper applies to virtually all symmetric functions with finite radius of convergence. We show examples in machine learning that respect permutation symmetry.

(i) **Data symmetry.** The loss function we minimize is $L(\theta, \{(x_i, y_i)\}_{i=1}^d)$, where $\theta$ stands for the trainable parameters, and $(x_i, y_i)$ is a data point consisting of input–label pairs. The loss function is the average of a per-sample loss $\ell$ over training data: $L = \frac{1}{d} \sum_{i=1}^d \ell(x_i, y_i, \theta)$. Permuting any pair of these data points results in the same loss function for any $\theta$.

(ii) **Neuron symmetry.** Consider a two-layer neural network $f(x) = W_2 \sigma(W_1 x)$, where $W_1$ and $W_2$ are weight matrices, and $\sigma$ is an activation function. Let $w_i^T$ be the row vectors of $W_1$, and $v_i$ be the column vectors of $W_2$. Then the output reads $f(x) = \sum_{i=1}^d v_i \sigma(w_i^T x)$, where $d$ is the width of the hidden layer. The output is symmetric under the exchange of any pair of $(v_i, w_i) \leftrightarrow (v_j, w_j)$ (see Fig. 1(b)). Other modules that have permutation symmetry include fully connected layers, attention logits in self-attention, and attention outputs between different attention heads (Brea et al., 2019; Ziyin et al., 2025). Particularly, we elaborate on the permutation symmetry in attention modules in Appendix H.

## 2   UNIVERSAL COMPRESSION THEOREM

The value of a symmetric function $f(\theta)$ only depends on where $\theta = (w_1, \dots, w_d)$ lives in the quotient space $V^d / S_d$. When $d$ is large, the distribution of $w_i$'s likely becomes crowded and the representation becomes redundant. We show that this redundancy can be removed by replacing $\theta$ with a much smaller *weighted* subset that preserves all relevant symmetric evaluations up to vanishing error. We defer the comprehensive theory to Appendix.

The first fact to notice is that the value of $f(\theta)$ is entirely determined by the (tensorial) *statistical moments*:

$$p_k = \frac{1}{d} \sum_i w_i^{\otimes k} \equiv \frac{1}{d} \sum_i \underbrace{w_i \otimes \dots \otimes w_i}_{k \text{ repetitions}}. \tag{2}$$

Let us consider $f(\theta)$ that is Taylor-expandable. Expanding $f$ to order $k$ yields a symmetric polynomial of degree $k$. We can match these moments with much fewer objects, and it follows that the error starts from the $(k+1)$th order. Specifically, it is possible to match the first $k$ moments $p_1, \dots, p_k$ with only finitely (i.e., not scaling up with $d$) many reweighted objects. This is guaranteed by Tchakaloff's theorem (Tchakaloff, 1957), and is algorithmically executable by Algorithm 2.

We propose a family of compression algorithms (Algorithm 1). The overall strategy is that, in each iteration, one finds a subset of objects within a small diameter ($\sim d^{-1/m}$); then one replaces these objects by fewer objects with modified weights while preserving the first $k$ moments. We prove that (see Appendix E for the precise statement and proof), with this compression strategy, one can compress a $d$-object set into $\mathrm{polylog}(d)$ with vanishing error:

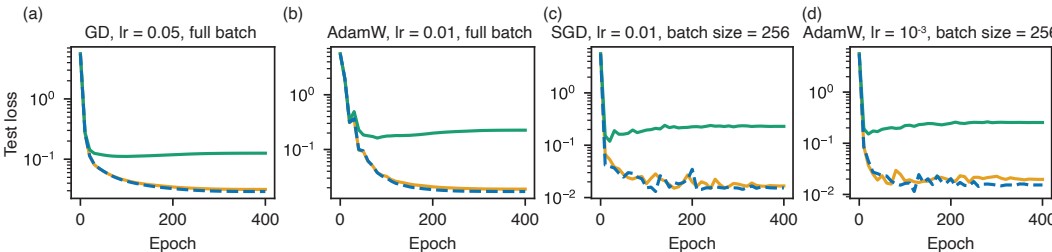

Figure 2: Compression of the training dataset in a teacher–student setup. Details described in Appendix G. Green dashed line: training with the original dataset of size $d = 10^4$; Orange line: training with a compressed dataset of size $10^3$, using order $k = 5$ moment matching. Blue line: training with a size-$10^3$ subset of the original dataset.

**Theorem 1** (Universal compression; informal statement). *Assume $\|w_i\| \leq R$ and $f$ is permutation invariant and analytic with radii of convergence independent of $d$. Then for any target error $\varepsilon(d) \to 0$, there exists a weighted subset $\theta'$ supported on*

$$d' = O\left(\left(\log \frac{d}{\varepsilon(d)}\right)^m\right) \tag{3}$$

*objects, such that $|f(\theta') - f(\theta)| \leq \varepsilon(d)$. Moreover, this rate is optimal up to constants: there exist $d$-object configurations for which no universal method can compress to $o(\log^m d)$ weighted objects while preserving all symmetric functions.*

**Numerical simulation.** Figure 2 is our first validation of the moment-matching compression. We train a simple neural network with a large ($10^4$) dataset. A much smaller ($10^3$) compressed dataset achieves almost the same training dynamics, whereas another independently sampled dataset of the same size performs differently.

## 3    DYNAMICAL LOTTERY TICKET HYPOTHESIS

The lottery ticket hypothesis (LTH) (Frankle & Carbin, 2018) postulates that within any sufficiently wide neural network, one can find a subnetwork that, when trained in isolation, achieves performance comparable to the original. While widely observed, its theoretical understanding has remained elusive. As a corollary of the universal compression theorem, here we establish that any layer of a neural network with width $d$ can be compressed losslessly to a $\mathrm{polylog}(d)$ size, such that the training dynamics before and after compression are identical. We refer to this statement as the *dynamical LTH*, which can be viewed as a stronger and more quantitative variant of the original hypothesis, which postulates that the final results are almost identical.

The dynamical LTH is rigorously presented in Theorem 9 of Appendix F. The key idea is that predictions and loss functions are not only symmetric functions of the trained parameters, but can also be regarded as symmetric functions of the initial parameters. This holds because common update rules are *equivariant*, i.e., they commute with permutations (see Appendix F). Let $f$ denote a symmetric function, let $\mathcal{T}$ denote the training dynamics (a mapping from initial parameters to trained parameters), and let $\theta_0$ denote the initial network parameters. By equivariance, $f(\mathcal{T}(\theta_0))$ is again a symmetric function of $\theta_0$. Therefore, the moment-matching compression established earlier applies directly to $f \circ \mathcal{T}$, leading to the dynamical LTH.

**Numerical simulation.** Figure 3 validates the dynamical LTH. The task is to learn the function $f(x_1, x_2)$ shown in Fig. 3(a) from $10^5$ data points of the form $(x_1, x_2, \mathcal{N}(f(x_1, x_2), 0.2^2))$, with $x_{1/2}$ sampled uniformly from $[-1, 1]$. Across a wide range of update rules and learning rates, the predictions of a wide network and its compressed counterpart are shown to be nearly indistinguishable throughout the training dynamics.

## 4    IMPROVING NEURAL SCALING LAWS

Theorem 1 can be leveraged to asymptotically improve neural scaling laws. A commonly used empirical form of neural scaling laws is $L(d) = L_0 + (d_0/d)^\alpha$, where $L$ denotes the loss and $d$ may represent dataset size, number of parameters, or similar resources (Henighan et al., 2020). If we

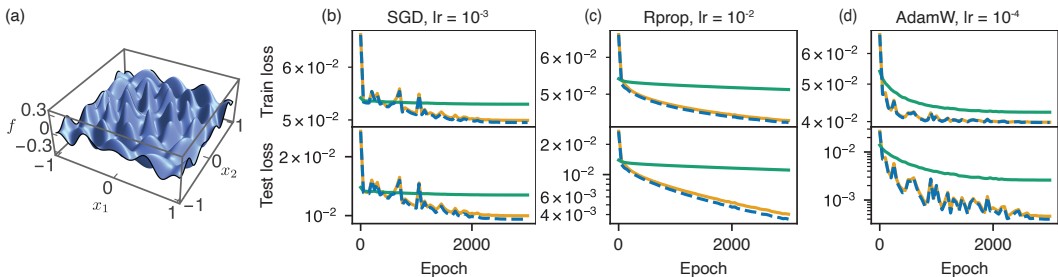

Figure 3: Dynamical LTH (Theorem 9). Details of the task are described in Appendix G. (a) Ground-truth function $f(x_1, x_2) = J_6(20r)\cos(6\theta)$, where $r^2 = x_1^2 + x_2^2$ and $\theta = \arctan(x_2/x_1)$, known as a cylindrical harmonic. (b–d) MSE loss vs epoch under three different update rules. Green dashed line: randomly initialized network of width $10^4$; Orange line: compressed network of width $10^3$, using $k = 5$ moment matching; Blue line: random subnetwork of the $10^4$-width network, also of width $10^3$.

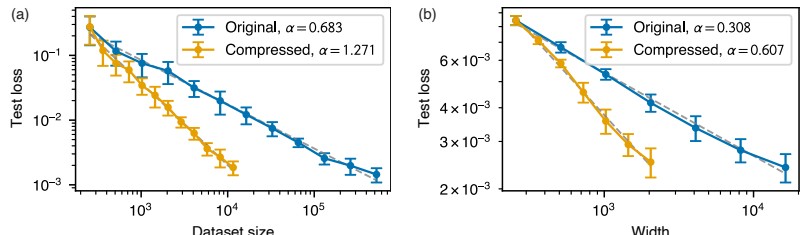

Figure 4: Improving neural scaling laws through compression. (a) MSE loss of the teacher–student task after training on an original dataset of size $d$ vs a compressed dataset of size $d'$. (b) MSE loss of the cylindrical harmonic task after training a two-layer neural network of width $d$ versus its compressed counterpart of width $d'$. In both panels, we compress $d$ objects to $d' = \lceil 16\sqrt{d} \rceil$ using $k = 6$ moment matching. The exponent $\alpha$ is obtained by fitting $L \propto d^{-\alpha}$ or $d'^{-\alpha}$.

compress $d$ to $d' = O(\log^m d)$ (inducing arbitrarily fast power law error, which is negligible), any power-law scaling will be improved to a stretched exponential scaling as

$$L(d') = L_0 + d^{-\alpha} = L_0 + \exp\left(-\alpha' \sqrt[m]{d'}\right). \tag{4}$$

With a more efficient compression algorithm (see Theorem 6), we may also compress $d$ to $d' = O(d^\sigma)$, where $0 < \sigma < 1$, so that the neural scaling exponent $\alpha$ is divided by $\sigma$—a smaller but computationally manageable improvement on neural scaling laws.

**Numerical simulation.** We demonstrate improvements in scaling laws with respect to dataset size in Fig. 4(a) and with respect to network width in Fig. 4(b). The learning tasks in Figs. 4(a) and 4(b) are the same as those in Figs. 2 and 3, respectively; additional details are provided in Appendix G. In both cases, compressing $d$ objects to $\lceil 16\sqrt{d} \rceil$ objects effectively doubles the scaling exponent.

## 5 CONCLUSION AND OUTLOOK

We introduced a system-agnostic framework proving that any permutation-symmetric function admits strong compression without sacrificing performance, yielding a unified perspective on the compressibility of both models and datasets. The central contribution of our work is a proof of concept that it is theoretically possible to strongly compress neural networks and datasets, enabling far more efficient use of data and parameters.

The main future direction is to improve the practicality of universal compression. Our compression procedure is currently too slow and memory-intensive in high data dimensions; we expect future works to optimize it or derive accurate, scalable approximations. A second direction is to extend the framework to high ambient dimension by exploiting structure: many real-world objects concentrate near low-dimensional embeddings (Abbas et al., 2021), and language in particular appears to have small effective dimension (Gromov et al., 2023). Finally, the theory suggests that there exist better sampling and initialization schemes that are "compressed by construction," potentially connecting

to importance sampling (Hammersley, 2013; Bengio & Senecal, 2008) and orthogonal initialization (Saxe et al., 2014).

## ACKNOWLEDGMENT

The authors thank Yizhou Xu for discussions. ILC acknowledges support in part from the Institute for Artificial Intelligence and Fundamental Interactions (IAIFI) through NSF Grant No. PHY-2019786. This work was also supported by the Center for Brains, Minds and Machines (CBMM), funded by NSF STC award CCF-1231216.

## REPRODUCIBILITY STATEMENT

A realization of our compression algorithm and code for generating all the figures are available at `https://github.com/WHY-David/moment_compression.git`.

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

## A NOTATIONS AND CONVENTIONS

Let $V = \mathbb{R}^m$ denote the space in which each object $w_i$ is embedded. $\otimes$ represents tensor product. The shorthand $[d]$ refers to the index set $\{1, 2, \ldots, d\}$, but when $x$ is a non-integer real number, $[x]$ denotes its closest integer. $S_d$ denotes the symmetric group on $d$ elements. For a nonnegative weight vector $\{c_i\}_{i=1}^d$, the support is defined as $\mathrm{supp}(c_i) \equiv \{i \in [d] \mid c_i \neq 0\}$. For a set $S \subseteq V$, the diameter is $\mathrm{diam}(S) \equiv \max_{x, x' \in S} \|x - x'\|$, where we use the Euclidean norm throughout this paper. We write $\mathcal{N}(\mu, \sigma^2)$ for the normal distribution with mean $\mu$ and variance $\sigma^2$. All other notations will be introduced in context.

We use $\theta = (w_1, \ldots, w_d)$ as a collective notation for all objects. To analyze the error induced by compression, we impose a mild regularity assumption on $f$. Specifically, it is known that any symmetric function admits a "deep set"–style universal representation (Zaheer et al., 2018) of the form

$$f(w_1, \ldots, w_d) = h\left(\sum_{i=1}^{d} g(w_i)\right), \tag{5}$$

where $h$ and $g$ are suitable functions. Importantly, for smooth $f$, one can choose $h$ and $g$ to be smooth as well (Tabaghi & Wang, 2023). We will use the following regularity assumption for the symmetric functions considered in this paper.

**Assumption 1.** *For all symmetric functions $f(\theta)$ studied in this paper, we assume that there exists a deep–set representation of $f$ as in Eq. (5), such that*

1. *Neither $h$ nor $g$ depends on $d$;*

2. *$h$ and $g$ are both Taylor-expandable with finite radii of convergence.*

While many ML models are non-smooth (e.g., ReLU networks), our compression results often extend to such settings at the level of conclusion (concretely, ReLU networks are studied in the numerical results in Figs. 2, 3 and 4), suggesting broader applicability beyond the analytic regime treated formally here.

## B    RELATED WORKS

**Compression in AI.**    Model and dataset compression has long been a central problem in AI (Han et al., 2015; Frankle & Carbin, 2018; Sorscher et al., 2022; Salomon, 2002; Wang et al., 2018). Yet, almost no theoretical framework exists to explain why, or to what extent, such compression is possible. A primary conceptual framework is the lottery ticket hypothesis (LTH) (Frankle & Carbin, 2018), which posits that within every network there exists a small subnetwork that, when retrained, can achieve the same performance as the original. Several theoretical works have established variants of the LTH (Malach et al., 2020; Pensia et al., 2021; da Cunha et al., 2022). However, these results typically fail to imply that the compressed model exhibits the same learning dynamics as the original—that is, that it reaches the same performance after training—which is arguably the most practical implication of the LTH. To date, the original formulation of the LTH remains unproved, precisely because of its dual requirement of both training and compression. We provide a more detailed discussion of this point in Sec. 3. Another closely related work is Ziyin (2024), which suggests the connection between symmetries and emergent sparsity during training.

**Neural Scaling Laws.**    A major empirical guideline for training large language models (LLMs) is the neural scaling laws, which state that as the size of models and datasets increases, the generalization error decays as a power law: $L \propto d^{-\alpha}$, with $\alpha$ often small (Kaplan et al., 2020). Such small exponents pose a central obstacle for scaling LLMs. For example, when $\alpha = 0.1$, reducing the generalization error by half would require increasing the dataset size by a factor of $1000$—an impractical demand given today's limited data availability. Sorscher et al. (2022) suggest the possibility of improving scaling laws through data pruning; however, their theory applies only to linear regression and assumes knowledge of the ground-truth model. Whether scaling laws can be improved in more general settings, and without requiring access to the ground truth, remains unknown.

## C    PRELIMINARIES FOR THE UNIVERSAL COMPRESSION THEOREM

### C.1    MULTIVARIATE FUNDAMENTAL THEOREM OF SYMMETRIC POLYNOMIALS

Here, we prove the fact asserted at the beginning of Sec. 2: the value of a symmetric polynomial is completely determined by the tensorial statistical moments. It is formally stated as Theorem 3.

We first quote the following well-known theorem without proof.

**Theorem 2** (Fundamental theorem of symmetric polynomials (FTSP) (Macdonald, 1998))**.** *For any symmetric polynomial $f(x_1, \ldots, x_d)$, where $x_i \in \mathbb{R}$ for all $i \in [d]$, there is a unique polynomial $P$*

*such that*

$$f(x_1, \ldots, x_d) = P(p_1, \ldots, p_d), \tag{6}$$

*where $p_k$ ($k \in [d]$) is called the kth moment and is defined as*

$$p_k = \frac{1}{d} \sum_{i=1}^{d} x_i^k. \tag{7}$$

An arguably more common statement of the FTSP is that any symmetric function is a function of power sums $\sum_{i=1}^{d} x_i^k$ for $k = 0, 1, \ldots, d$, which differs from our statement by a normalization of $d$. However, our convention fixes each moment $p_k$ to constant order of magnitude. With our convention, we can treat $P$ as dependent on $d$.

**Theorem 3** (FTSP, Multivariate Variant). *Let $\theta = (w_1, \ldots, w_d)$ with $w_i \in \mathbb{R}^m$, and $f(\theta)$ be a polynomial in all scalar components $w_{i,a}$. Then, any symmetric function $f(\theta)$ can be expressed as a function of the moments $p_k$, $k \in [d]$, defined by*

$$p_k = \frac{1}{d} \sum_i w_i^{\otimes k} \equiv \frac{1}{d} \sum_i \underbrace{w_i \otimes \cdots \otimes w_i}_{k \text{ repetitions}}. \tag{8}$$

*Proof.* We write the coordinates of $w_i$ as $(w_{i,1}, \ldots, w_{i,m})$. For convenience, we use the multi-index notation $\alpha = (a_1, \ldots, a_m)$, such that $w_i^\alpha \equiv w_{i,1}^{a_1} \ldots w_{i,m}^{a_m}$. For a multi-index $\alpha$, $|\alpha| \equiv a_1 + \cdots + a_m$. Also, we define $q_\alpha = \sum_{i=1}^{d} w_i^\alpha$ (we use $q$ to distinguish from $p$ with a $1/d$ factor). Our proof is divided into a few steps: (1) Represent $f$ in terms of $q_\alpha$'s (2) Repack $\{q_\alpha\}$ into tensor moments $\{p_k\}$ (3) Show that only $p_k$ for $k = 1, \ldots, d$ are independent (4) Show that the representation in terms of $\{p_k\}$ is unique.

Generally, a polynomial $f$ is linearly spanned by monomials in the form $w_{i_1}^{\alpha_1} w_{i_2}^{\alpha_2} \ldots w_{i_N}^{\alpha_N}$ (note that $N < d$), where all subscripts are distinct and each $|\alpha_j| > 0$. Imposing permutation symmetry $S_d$, such monomials differing by index permutation must appear with the same coefficient. That is, the space of symmetric polynomials is linearly spanned by terms in the form

$$\sum_{\sigma \in S_d} w_{\sigma(1)}^{\alpha_1} w_{\sigma(2)}^{\alpha_2} \cdots w_{\sigma(N)}^{\alpha_N} \propto \sum_{\mathcal{X}(i_1, \ldots, i_N)} w_{i_1}^{\alpha_1} w_{i_2}^{\alpha_2} \ldots w_{i_N}^{\alpha_N} \tag{9}$$

Here, we introduced the notation $\mathcal{X}(i_1, \ldots, i_N)$ to denote the set of index list $(i_1, \ldots, i_N)$ such that all $i_j$ are distinct and range from 1 to $d$.

We prove by induction on $N$ that such terms can all be written as polynomials of $q_\alpha$'s. When $N = 1$, Eq. (9) is simply $q_\alpha$ by definition:

$$\sum_{i=1}^{d} w_i^\alpha = q_\alpha. \tag{10}$$

Now, suppose Eq. (9) is proved to be a polynomial of $q_\alpha$'s. Replacing $N \to N + 1$, we look at

$$\sum_{\mathcal{X}(i_1, \ldots, i_{N+1})} w_{i_1}^{\alpha_1} w_{i_2}^{\alpha_2} \ldots w_{i_{N+1}}^{\alpha_{N+1}}. \tag{11}$$

We use the set decomposition

$$\mathcal{X}(i_1, \ldots, i_N) \times \{i_{N+1}\}_1^d = \mathcal{X}(i_1, \ldots, i_{N+1}) \sqcup \mathcal{X}(i_1, \ldots, i_N) \times \{i_1\}$$
$$\sqcup \cdots \sqcup \mathcal{X}(i_1, \ldots, i_N) \times \{i_N\} \tag{12}$$

to rewrite Eq. (11) as

$$\sum_{\substack{\mathcal{X}(i_1, \ldots, i_N) \\ i_{N+1}}} w_{i_1}^{\alpha_1} w_{i_2}^{\alpha_2} \ldots w_{i_{N+1}}^{\alpha_{N+1}} = \sum_{\mathcal{X}(i_1, \ldots, i_N)} w_{i_1}^{\alpha_1} w_{i_2}^{\alpha_2} \ldots w_{i_{N+1}}^{\alpha_{N+1}}$$
$$- \sum_{\mathcal{X}(i_1, \ldots, i_N)} w_{i_1}^{\alpha_1} w_{i_2}^{\alpha_2} \ldots w_{i_N}^{\alpha_N} w_{i_1}^{\alpha_{N+1}} - \cdots - \sum_{\mathcal{X}(i_1, \ldots, i_N)} w_{i_1}^{\alpha_1} w_{i_2}^{\alpha_2} \ldots w_{i_N}^{\alpha_N} w_{i_N}^{\alpha_{N+1}}. \tag{13}$$

On the right-hand side, the first term can be written as

$$\left( \sum_{\mathcal{X}(i_1, \ldots, i_N)} w_{i_1}^{\alpha_1} w_{i_2}^{\alpha_2} \ldots w_{i_N}^{\alpha_N} \right) q_{\alpha_{N+1}}, \tag{14}$$

in which the parenthesis is a polynomial of $q_\alpha$'s by induction assumption. The second term can be written as

$$\sum_{\mathcal{X}(i_1,\ldots,i_N)} w_{i_1}^{\alpha_1+\alpha_{N+1}} w_{i_2}^{\alpha_2} \ldots w_{i_N}^{\alpha_N}, \tag{15}$$

which is a polynomial of $q_\alpha$'s as well, and so are the other terms in the ellipse. Therefore, we proved that all terms in the form of Eq. (9) can be written as polynomials of $q_\alpha$'s.

Next, we repack $q_\alpha$ into tensors. Define $p_\alpha = q_\alpha/d$. Note that the set $\{p_\alpha \mid |\alpha| = k\}$ is exactly the set of coordinates of the tensor $p_k = \sum_i w_i^{\otimes k}/d$. So indeed, the value of $f$ relies only on the tensor moments $p_k$.

To show that $f$ only requires the first $d$ moments, we try to represent any $p_k$ ($k > d$) as a function of $p_1, \ldots, p_d$. $p_k$ is a symmetric tensor of rank $k$. It is easy to see that the space of symmetric $k$-tensors is isomorphic to the space of homogeneous polynomials of degree $k$ (the argument is denoted as $u \in \mathbb{R}^m$), by the homomorphism

$$T_{i_1,\ldots,i_k} \to \sum_{i_1,\ldots,i_k} T_{i_1,\ldots,i_k} u_{i_1} \ldots u_{i_k}. \tag{16}$$

Specifically, $p_k$ is mapped to

$$s_k(u) = \frac{1}{d} \sum_{i=1}^N (u^\top w_i)^k \tag{17}$$

Using Theorem 2 for the variables $\{u^\top w_i \mid i = 1, 2, \ldots, d\}$, there is a polynomial $P$ such that

$$s_k(u) = P(s_1(u), \ldots, s_d(u)). \tag{18}$$

Because this holds for arbitrary $u$, it follows that $p_k$ is a function of $p_1, \ldots, p_d$ as well.

The fact that the representation of $f$ in terms of moments is unique is obvious: assume two polynomials $f$ and $f'$ are mapped to the same function. Taking the difference of the two equations $f(\theta) = P(\{p_k\})$ and $f'(\theta) = P(\{p_k\})$, we find $f(\theta) - f'(\theta) = 0$ for any $\theta$. □

### C.2 REDUCING SUPPORT WHILE MATCHING MOMENTS

**Theorem 4** (Tchakaloff (1957)). *Let $\mu$ be a measure supported on $D \subset \mathbb{R}^m$. Then there exist $N$ points $w_j \in D$, with $N \le N_{m,k} = \binom{m+k}{k}$, and positive weights $c_j$, such that the first $k$ moments are matched:* $\forall l \in \{0, 1, \ldots, k\}, \int_D w^{\otimes l} d\mu(w) = \sum_{j=1}^N c_j w_j^{\otimes l}.$

A proof follows from Carathéodory's theorem in dimension $N_{m,k}$ (Leonard & Lewis, 2015). Note that $N_{m,k}$ is precisely the dimension of the linear space of all moments up to order $k$ (including one fictitious dimension corresponding to $p_0$). Algorithm 2 in Appendix D.1 guarantees such a compression: whenever there are more than $N_{m,k}$ weighted objects, we can always reduce the support to at most $N_{m,k}$ objects while preserving the first $k$ moments.

In the spirit of Tchakaloff's theorem, one can compress the original parameter set $\theta$ into a smaller set of weighted parameter set $\theta'$:

**Definition 1.** *A weighted parameter set $\theta'$ is defined as a collection of weight-parameter pairs*

$$\theta' = \{(c_1, w_1), (c_2, w_2), \ldots, (c_{d'}, w_{d'})\}, \tag{19}$$

*where each $c_j \ge 0$ and $w_j \in V = \mathbb{R}^m$. The moments of $\theta'$ and the values of symmetric functions are defined as if there exist $c_j$ copies of $w_j$:*

$$p_k' = \frac{1}{\sum_{j=1}^{d'} c_j} \sum_{j=1}^{d'} c_j w_j^{\otimes k}, \quad f(\theta') = \rho\left(\sum_{j=1}^{d'} c_j g(w_j)\right). \tag{20}$$

We remark that the original $\theta$ can also be viewed as weighted, with unit weight for each object. A feature of our compression is that it does not change any of the $w_i$'s but instead adjusts the weights so that they are supported on a smaller subset. When $f$ and $\{w_i\}_{i\in[d]}$ are fixed, we may regard the output as a function only of the weights $\{c_i\}_{i\in[d]}$. Specifically, if $f_d(\theta) = \rho(\sum_i g(w_i))$, we denote $\phi(c) = \rho\left(\sum_{i=1}^d c_i g(w_i)\right)$.

We are now ready to study approximating symmetric functions $f$ that are Taylor-expandable. Expanding $f$ to order $k$ yields a symmetric polynomial of degree $k$, and these moments can be matched with very few weighted objects. Consequently, the approximation error begins at order $(k+1)$.

**Theorem 5** (Moment matching in a small ball). *Let $f$ be a symmetric function acting on a weighted parameter set $\theta = \{(c_i, w_i)\}_{i \in [d]}$, and let $\phi$ be its corresponding function acting on weights. Let $r = \max_{i \neq j} \|w_i - w_j\|$. Then, there exists a $\theta' = \{(c_i', w_i)\}_{i \in [d]}$ such that no more than $N_{m,k} \equiv \binom{m+k}{k}$ weights are nonzero, and*

$$|\phi(c') - \phi(c)| = O(dr^{k+1}). \tag{21}$$

*Proof.* Choose an $m$-dimensional ball of diameter $r$ that contains all $w_i$'s, and let the center be $w_0 \in \mathbb{R}^m$. We first Taylor-expand $f(\theta)$ around $\theta_0 = (w_0, \ldots, w_0)$ up to the $k$th order. The expansion is a symmetric polynomial of degree $k$, so it can be written as a function of $(p_1, \ldots, p_k)$. By Theorem 4, we can find another set of weights $\{c_i'\}$ supported on $N_{m,k} = \binom{m+k}{k}$ points, such that

$$\sum_i c_i'(w_i - w_0)^{\otimes l} = \sum_i c_i(w_i - w_0)^{\otimes l} = p_l \quad \text{for } l = 1, \ldots, k \tag{22}$$

Let $\theta' = \{c_i', w_i\}$, and denote its moments by $\{p_l'\}$. By construction, we have $p_l' = p_l$ for $l = 1, \ldots, k$. Since the first $k$ moments all match, there is no difference between $\phi(c)$ and $\phi(c')$ up to the $k$th order.

Next, we look at the $(k+1)$th order in the Taylor expansion of $f(\theta)$ around $\theta_0$. It is written as

$$\sum_{|\alpha|=k+1} \frac{1}{\alpha!} \partial_\alpha f(\theta_0)(\theta - \theta_0)^\alpha \tag{23}$$

It is a degree-$(k+1)$ homogeneous symmetric polynomial, and we denote it as $P_{k+1}(p_1, \ldots, p_{k+1})$. Hence,

$$f_d(\theta) - f_d(\theta') = P_{k+1}(p_1, \ldots, p_{k+1}) - P_{k+1}(p_1, \ldots, p_k, p_{k+1}'). \tag{24}$$

In $P_{k+1}(p_1, \ldots, p_{k+1})$, the only term that depends on $p_{k+1}$ is linear in itself—$\langle b_{k+1}, p_{k+1} \rangle$, where $b_{k+1}$ is a $(k+1)$-index symmetric coefficient tensor and here $\langle \cdot, \cdot \rangle$ denotes tensor contraction; all other terms are completely determined by $\{p_l\}_{l=1}^k$. To see that $b_{k+1}$ is at most of order $d$, we use the deep-set–representation: $f(\theta) = \rho(\sum_{i \in [d]} g(w_i))$. Then $b_{k+1}$ can be written down explicitly as

$$\begin{aligned}
\langle b_{k+1}, p_{k+1} \rangle &= \frac{\partial f}{\partial y} \sum_{i=1}^d \left\langle a_{k+1}, (w_i - w_0)^{\otimes(k+1)} \right\rangle \\
&= d \left\langle \frac{\partial f}{\partial y} a_{k+1}, p_{k+1} \right\rangle,
\end{aligned} \tag{25}$$

where $y = \sum_{i \in [d]} g(w_i)$, and $\langle a_{k+1}, (w_i - w_0)^{\otimes(k+1)} \rangle$ is the $(k+1)$th order in the Taylor expansion of $g$ around $w_0$. All derivatives are taken at $\theta = \theta_0$. By our convention that $\rho$ and $g$ are independent of $d$, we thus have $b_{k+1} = O(d)$.

Also, since each $\|w_j - w_0\| \leq r/2$ we have $p_{k+1} = O(r^{k+1})$. Therefore, we conclude that $f_d(\theta) - f_d(\theta') = O(dr^{k+1})$. $\qquad\qquad\square$

Theorem 5 is a main ingredient of our compression theory. A lesson from this theorem is that dealing with a group of objects of small diameter is very effective in suppressing error, as we can choose a large enough $k$ to greatly suppress the compression error.

### C.3 Sphere packing lemma

This lemma is used in proving Theorems 6 and 7.

**Lemma 1** (Sphere packing). *Given $d \gg 1$ objects in a closed $m$-dimensional ball of radius $R$: that is, $\theta = \{w_i\}_{i=1}^d$, $\|w_i\| \leq R$. For any $\theta$, the diameter of the smallest ball containing $N$ points is at most of order $(N/d)^{1/m}$. That is,*

$$\sup_\theta \min_{\substack{S \subset \theta \\ |S|=N}} \text{diam}(S) = R\, O\left((N/d)^{1/m}\right). \tag{26}$$

*Proof.* We use $B(x, r)$ to denote a closed $m$-ball centered at $x$.

Cover $B(0, R)$ by $M$ balls of radius $r$. By the pigeonhole principle, if $d > (N-1)M$, some ball contains at least $N$ points, giving an $N$-point subset of diameter $\leq 2r$.

We show that there exists a covering with $M = (1 + 2R/r)^m$ points. We choose a set of points $\{x_j\}_{j=1}^{M} \subset B(0, R)$ to form a maximal $r/2$-packing of the ball $B(0, R)$, meaning

$$\|x_i - x_j\| \geq r, \forall i \neq j, \tag{27}$$

and no further points can be added while maintaining this separation. By this definition, the balls $\{B(x_j, r)\}$ form a covering of $B(0, R)$, but the smaller balls $\{B(x_j, r/2)\}$ are mutually disjoint and contained in $B(0, R + r/2)$. By comparing volumes, we find

$$M(r/2)^m \leq (R + r/2)^m. \tag{28}$$

Hence, if the radius of each ball is $r$, there exists a covering of $B(0, R)$ with $\lceil (1 + 2R/r)^m \rceil$ balls. Also requiring $d/(N-1) > \lceil (1 + 2R/r)^m \rceil$, we conclude that

$$\sup_{\theta} \min_{\substack{S \subset \theta \\ |S|=N}} \operatorname{diam}(S) \leq \frac{4R}{\left(\frac{d}{N-1} - 1\right)^{1/m} - 1} = R\, O((N/d)^{1/m}). \tag{29}$$

$\square$

## D  A COMPRESSION ALGORITHM FAMILY

Theorem 5 shows that compressing points in a small diameter enables a tight control on the error. By a sphere packing argument (see Lemma 1), if we put many ($d$) objects in a finite region, then we can always find a subset of diameter $O((N/d)^{1/m})$ containing $N$ objects. This gives rise to a general strategy of compression in Algorithm 1, and any concrete algorithm that fits in this strategy is guaranteed to have vanishing compression error.

---

**Algorithm 1:** A compression algorithm family.

---
**Input**  : a set of objects $\theta = \{w_i\}_{i \in d}$, target size $d'$, moment-matching order $k \in \mathbb{Z}_+$
**Output:** $\theta' = \{c_j, w_j\}_{j \in \operatorname{supp}(c)}$, where $|\operatorname{supp}(c)| \leq d'$
Initialize $c_i = 1$ for all $i \in [d]$;
**while** $|\operatorname{supp}(c)| > d'$ **do**
  **Step 1 (clustering)**: Find a cluster $S \subseteq \operatorname{supp}(c)$ such that $|S| > N_{m,k}$ and
  $\operatorname{diam}\{w_j\}_{j \in S} = O(d^{-1/m})$ ;
  **Step 2 (moment matching)**: Update weights $\{c_j\}_{j \in S}$ such that they are supported on $N_{m,k}$
  objects, and the first $k$ moments are kept unchanged.
**end**

---

In terms of a practical compression algorithm, we first remark that Step 2 (moment matching) can be realized by Algorithm 2 in Appendix D.1, although several other moment-matching algorithms exist, especially in the case of peeling many points in one cluster (Piazzon et al., 2017). The most time-consuming part in Algorithm 2 is finding null vectors, so we estimate its time complexity as $dN_{m,k}^2$, which tells us that matching higher moments takes a much longer time. For clustering, ideally in each iteration one wants to greedily find the $(N_{m,k} + 1)$-subset with the smallest diameter, but the $k$-nearest neighbor problem is known to be NP hard in general. For clarity, in Theorem 6 of this section, we prove error bounds associated with the greedy strategy, indicating that a compression map satisfying our asserted error bounds universally exists. However, we perform numerical experiments using $k$-means clustering (see Appendix D.1), which is much faster in practice.

### D.1  AN EXAMPLE OF THE MOMENT MATCHING ALGORITHM

Here, we describe the concrete algorithm for compression that is used in all our numerical simulations.

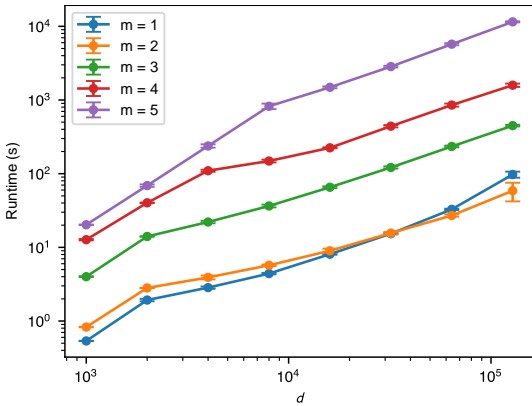

Figure 5: Runtime benchmark of the hybrid compression algorithm. See description in Sec. D.1. The original dataset is i.i.d. uniformly sampled in an $m$-dimensional cube. The moment-matching order is $k = 5$ for all trials in this plot. Each marker on the plot is an average over 5 trials, with error bar representing one standard deviation.

Recall that in Algorithm 1 we identified two main steps of the algorithm: (1) clustering and (2) moment matching. We first describe our moment matching strategy. For moment-matching, we consistently use Algorithm 2, in which $\text{vect}(\cdot)$ represents flattening all the moments into a vector. We remark that the existence of Algorithm 2 is effectively a constructive proof of Tchakaloff's theorem 4.

Our actual clustering strategy is slightly more involved, because finding the smallest cluster in $d \gg 1$ points is known to be NP hard. We implement a coarser $k$-means clustering instead. Only when $|\text{supp}(c)|$ becomes close to the desired stopping size $d'$, we switch to a greedy strategy, since the diameters of clusters are likely to be large when $|\text{supp}(c)|$ is small. Concretely, in a $k$-means round, we divide $\theta$ into $\propto |\text{supp}(c)|/N_{m,k}$ clusters. Then we apply Algorithm 2 to each cluster containing more than $N_{m,k}$ objects in parallel. In a greedy round, we find the approximately smallest cluster of $N_{m,k} + 1$ points, which is implemented using the $k$-nearest neighbor algorithm provided by the `faiss` package Douze et al. (2024).

For the above $k$-means/greedy hybrid strategy, we present the runtime benchmark in Fig. 5. The calculation is conducted on a personal computer with Apple M4 Pro CPU. The runtime is observed to be roughly proportional to $d$. This is because the number of iterations over moment-matching reduction of support is proportional to $d$.

Finally, we remind the reader that our error bound Theorems 6 and 7 are proved for the greedy strategy, where in each round one finds the smallest cluster of $N_{m,k} + 1$ objects and reduce one object. For the above-mentioned hybrid clustering strategy, there is no theoretical guarantee as strong as Theorem 6 for the error, but all numerical simulation turns out to meet our expectation.

---

**Algorithm 2:** Reducing support while matching moments

**Input** : Moment matching order $k$

**Input** : $N$ weighted parameters $\{(c_j, w_j)\}_{j=1}^N$, where $N_{m,k} = \binom{m+k}{k}$

**Output:** Adjusted weights $\{c_j\}$ where $|\text{supp}(c)| \leq N_{m,k}$

**function** $\phi(w) = \text{vect}(1, w, w^{\otimes 2}, \ldots, w^{\otimes k})^\top$    /* $\dim \phi(w) = N_{m,k}$       */;

**while** $|\text{supp}(c)| > N_{m,k}$ **do**

     Let $\text{supp}(c) = \{j_1, \ldots, j_{|\text{supp}(c)|}\}$; $A = \begin{pmatrix} \phi(w_{j_1}) & \phi(w_{j_2}) & \cdots & \phi(w_{j_{|\text{supp}(c)|}}) \end{pmatrix}$;

     Find a nonzero $v \in \mathbb{R}^{|\text{supp}(c)|}$ such that $Av = 0$; Ensure that at least one $v_j > 0$ ;

     $t = \min_{j \in \text{supp}(c): v_j > 0} c_j/v_j$;

     $c_j \leftarrow c_j - t v_j$.

**end**

---

# E    Universal compression theorems

## E.1    Compression error lower bound: finite $k$

The following theorem shows that with sufficiently large $k$ (compared to $m$, but still much smaller than $d$), one can compress the number of active objects to any power of $d$ with vanishing error.

**Theorem 6** (Universal Compression). *Let $\|w_i\| \le R$ for all $i \in [d]$. Algorithm 1 with moment-matching order $k$ can compress $\theta = \{w_i\}_{i=1}^d$ into $d' < d$ weighted objects $\theta'$, such that for any symmetric function $f$ satisfying Assumption 1,*

1. *The error is*

$$\mathcal{E} = |\phi(c') - \phi(c)| = O\left(d\,(d')^{1-\frac{k+1}{m}}\right), \tag{30}$$

 *where $\phi$ stands for the function $f$ treating weights as variables;*

2. *If $k > m - 1$, $d$ original objects can be compressed into*

$$d' = O\left(\left(\frac{d}{\varepsilon(d)}\right)^{\frac{m}{k-m+1}}\right) \tag{31}$$

 *weighted objects, such that the error is no larger than $\varepsilon(d)$.*

*Proof.* Denote $N_{m,k} = \binom{m+k}{k}$. In this proof, we show that the general algorithm (Algorithm 1) with greedy clustering strategy satisfies the asserted error bounds. By the greedy strategy (also described in Appendix D.1), we mean that in each iteration we choose a subset $S \subseteq |\mathrm{supp}(c)|$ of $N_{m,k} + 1$ objects among the active (i.e., with nonzero weight) objects. Then we reduce one object in $S$ while maintaining up to the $k$th moment.

By Theorem 5, the output error of one step is of order $c_S r^{k+1}$, where $c_S$ is the total weight of this cluster, and $r$ is the diameter. Because the greedy strategy always look for optimizing the diameter, after $O(d)$ steps, there is no better upper bound for $c_S$ than $d$. By Lemma 1, when there are $N$ active objects, the smallest diameter is upper-bounded by

$$O\left(\left(\frac{N_{m,k} + 1}{N}\right)^{1/m}\right) = O(N^{-1/m}). \tag{32}$$

Now, we sum up the error of all the steps. Denote the function's error as $\mathcal{E} = |\phi(c') - \phi(c)|$. Then, in one step of pruning, the error is

$$\Delta\mathcal{E} = O(dr^{k+1}) = O(dN^{-(k+1)/m}). \tag{33}$$

We upper-bound the sum over $N$ by an integral using $\sum_{N=d'+1}^d N^{-\alpha} \le \int_{d'}^d N^{-\alpha}\,dN$, and find that pruning $d$ original objects into $d'$ objects results in

$$\begin{aligned} \mathcal{E} &= O\left(\int_{d'}^d d\,N^{-(k+1)/m}\,dN\right) \\ &= O\left(\frac{d}{\frac{k+1}{m} - 1}\left((d')^{1-\frac{k+1}{m}} - d^{1-\frac{k+1}{m}}\right)\right) \end{aligned} \tag{34}$$

or logarithmic if $k + 1 = m$; but as we are only concerned with vanishing error, we will skip the analysis of $k + 1 = m$. Since $d' < d$, we conclude that the error is upper-bounded by

$$\mathcal{E} = O\left(d\,(d')^{1-\frac{k+1}{m}}\right), \tag{35}$$

which completes the proof of statement 1.

To show statement 2, we look at the equation

$$\varepsilon(d) = \mathcal{E} = O\left(d\,(d')^{1-\frac{k+1}{m}}\right) \tag{36}$$

Solving this equation with respect to $d'$ gives the asserted scaling of $d'$. $\qquad\square$

### E.2    Compression error lower bound: optimal $k$

In establishing Theorem 7, we have been fixing $k$ as a hyperparameter which can be chosen at will. Here, we will optimize over the choice of $k$ to study the smallest achievable final size $d'$ such that the error is vanishing. In the derivation below, $k$ could scale up with $d$, but we always set $m$ to be a finite constant.

**Theorem 7.** *Assume $\|w_i\| \le R$ for all $i \in [d]$. There exists a mapping from $d$ uniformly weighted objects to $d'$ weighted objects, such that for any symmetric function $f$ satisfying Assumption 1,*

1. *The error is*

$$\mathcal{E} = |\phi_d(\theta') - \phi_d(\theta)| = O\left(d(d')^{1-1/m} \exp\left[-\frac{1}{e}(m!\rho d')^{1/m}\right]\right), \tag{37}$$

   *where $\rho$ is the radius of convergence of the function $g$ in the deep-set representation of $f$ (Eq. (5));*

2. *$d$ uniformly weighted objects can be compressed into*

$$d' = O\left(\log \frac{d}{\varepsilon(d)}\right)^m \tag{38}$$

   *weighted objects, such that the error is no larger than $\varepsilon(d)$.*

*Proof.* Essentially, we follow the same greedy strategy and the same reasoning as the proof of Theorem 6. However, since $k$ can be large, here we keep track of all factors that scales with $k$ or $d$.

Recall that the upper bound for the radius of a cluster is $((N_{m,k} + 1)/N)^{1/m}$, and the upper bound for $c_S$ is $d$. Also, since the convergence radius of $g$ be $\rho$, the constant factor for the $(k+1)$th order Taylor expansion scales as $\rho^{-k}$. Putting these together, the error of one step of pruning is upper-bounded by

$$\Delta\mathcal{E} = O\left(d\rho^{-k}\left(\frac{N_{m,k} + 1}{N}\right)^{\frac{k+1}{m}}\right) \tag{39}$$

For notation simplicity, we will drop the big-$O$ notation and use $\Delta\mathcal{E}$ to refer to the upper-bound expression on the right-hand side of Eq. (39). There is an optimal $k_{\text{opt}}$ that minimizes the single-step error. We solve it by taking derivative of $\log(\Delta\mathcal{E}/d)$.

$$\begin{aligned}
\log(\Delta\mathcal{E}/d) &= -k\log\rho + \frac{k+1}{m}\left(\log(N_{m,k} + 1) - \log N\right) \\
&= -k\log\rho + \frac{k+1}{m}\left(\log(m+k)! - \log k! - \log(m!N)\right) + O(1) \\
&= k\log k + \log k - \frac{k+1}{m}\log(m!N\rho) + O(1).
\end{aligned} \tag{40}$$

In the third line, we used Stirling's formula $\log(n!) = n\log n - n + \frac{1}{2}\log(2\pi n) + \frac{1}{12n} + O(n^{-3})$ when $n \to \infty$ and simplified the expression. The derivative of the above reads

$$\frac{d}{dk}\log(\Delta\mathcal{E}/d) = \log k + 1 - \frac{1}{m}\log(m!N\rho) + O(k^{-1}). \tag{41}$$

Setting the derivative to zero, we obtain

$$\log k_{\text{opt}} = \frac{1}{m}\log(m!N\rho) - 1 + O(k^{-1}) \tag{42}$$

Plugging this value into Eq. (40), we get

$$\begin{aligned}
\log(\Delta\mathcal{E}_{\text{opt}}/d) &= -k_{\text{opt}} + O(1) \\
\Rightarrow \quad \Delta\mathcal{E}_{\text{opt}} &= O\left(d\exp\left[-\frac{1}{e}(m!N\rho)^{1/m}\right]\right).
\end{aligned} \tag{43}$$

Moving forward, we integrate over $\Delta \mathcal{E}_{\text{opt}}$ from $d'$ to $d$:

$$
\begin{aligned}
\mathcal{E}_{\text{opt}} &= O\left(d \int_{d'}^{d} \exp\left[-\frac{1}{e}(m!N\rho)^{1/m}\right] dN\right) \\
&= O\left(d\,\Gamma\left(m, \frac{1}{e}(m!\rho d')^{1/m}\right)\right),
\end{aligned}
\tag{44}
$$

where $\Gamma$ is the incomplete Gamma function:

$$
\Gamma(m, z) \equiv \int_{z}^{\infty} t^{m-1} e^{-t} dt.
\tag{45}
$$

Its asymptotic behavior at $z \to \infty$ reads

$$
\Gamma(m, z) = \exp\left[-z + O(z^{-1})\right] z^{m-1}\left(1 + O(z^{-1})\right).
\tag{46}
$$

Inserting this expansion into Eq. (44), we get the asserted error scaling in part 1 of the theorem.

To obtain part 2, we solve the inequality

$$
\varepsilon(d) = \mathcal{E} = O\left(d(d')^{1-1/m} \exp\left[-\frac{1}{e}(m!\rho d')^{1/m}\right]\right)
\tag{47}
$$

with respect to $d'$. One can take log on both sides of this inequality and safely neglect the factor $(d')^{1-1/m}$ to eventually get

$$
d' = O\left(\log \frac{d}{\varepsilon(d)}\right)^{m}.
\tag{48}
$$

$\square$

### E.3 polylog COMPRESSION RATE IS OPTIMAL

In this Appendix, we show that our algorithm of compressing $d$ objects into $O((\log d)^m)$ weighted objects is optimal up to a constant factor. The strategy to prove this is to find a $d$-point uniform distribution $\mu$, show that for any $d'$-point distribution $\mu'$ we can always find a function that evaluates to sufficiently distinct values on these two distributions.

What distribution is hard to compress? Inspired by the fact that we prioritize merging close-in-distance points in the main Algorithm 1, in an adversarial distribution, all points should be roughly equidistant from the neighbors, so there is no cluster that is particularly "easy" to compress. Hence, we introduce the following notion of quasi-uniformity:

**Definition 2** (Quasi-uniformity). *Let $D \subset \mathbb{R}^m$ be a fixed compact set with nonempty interior. A set $X_d = \{x_1, \ldots, x_d\} \subset D$ is quasi-uniform if there is a constant $C_D$ (independent of $d$) such that each Voronoi cell of $x_i$ has volume less than or equal to $\frac{C_D}{d}$.*

A quasi-uniform point set obviously exists: for example, it can be a maximal $O(d^{-1/m})$-packing of the region $D$ (see the proof of Lemma 1 for the definition).

Another lemma we are going to use is the following. The basic message is that a nontrivial degree-$k$ polynomial cannot be exponentially small on most of $D$; a set of finite measure must remain above $e^{-Ak}$.

**Lemma 2.** *Let $D \subset \mathbb{R}^m$ be a convex compact set, and let $|D|$ denote its volume. Let $p$ be a real polynomial on $D$ of degree $\leq k$ normalized by $\|p\|_{L^\infty(D)} \equiv \sup_{x \in D} |p(x)| = 1$. For any $t > 0$, define*

$$
S_t = \{x \in D : |p(x)| \geq t\}.
\tag{49}
$$

*Then*

$$
\frac{|S_t|}{|D|} \geq 1 - (t/2)^{1/k}.
\tag{50}
$$

*Proof.* We begin by quoting a multivariate Remez inequality (Theorem 1.2 in Brudnyi & Yomdin (2015)): for any measurable $E \subset D$ with $\lambda = \frac{|E|}{|D|} \in (0, 1]$ and any real polynomial $q$ of degree $\leq k$,

$$
\|q\|_{L^\infty(D)} \leq T_k\left(\frac{1 + (1-\lambda)^{1/m}}{1 - (1-\lambda)^{1/m}}\right) \|q\|_{L^\infty(E)},
\tag{51}
$$

where $T_k$ is the Chebyshev polynomial of the first kind.

For $t \in (0, 1)$, we use $E_t$ to denote the sublevel set:

$$E_t = \{x \in D : |p(x)| \leq t\}, \qquad \lambda_t = \frac{|E_t|}{|D|}. \tag{52}$$

Applying (51) with $q = p$, $E = E_t$ and using $\|p\|_{L^\infty(D)} = 1$ gives

$$1 \leq T_k\left(\frac{1 + (1 - \lambda_t)^{1/m}}{1 - (1 - \lambda_t)^{1/m}}\right) t. \tag{53}$$

Then, we make simplifications to the Chebyshev term in Eq. (53). For $x \geq 1$,

$$T_k(x) = \tfrac{1}{2}\left(x + \sqrt{x^2 - 1}\right)^k + \tfrac{1}{2}\left(x - \sqrt{x^2 - 1}\right)^k \geq \tfrac{1}{2}x^k. \tag{54}$$

Moreover, since $v \mapsto v^{1/m}$ is increasing and $v^{1/m} \geq v$ on $[0, 1]$,

$$\frac{1 + (1 - \lambda)^{1/m}}{1 - (1 - \lambda)^{1/m}} \geq \frac{1}{1 - (1 - \lambda)^{1/m}} \geq \frac{1}{\lambda} \qquad \text{for } \lambda \in (0, 1]. \tag{55}$$

Combining (54) and (55) in (53), we obtain

$$1 \leq \tfrac{1}{2} \lambda_t^{-k} t \quad \implies \quad \lambda_t \leq (t/2)^{1/k}. \tag{56}$$

Now we pass to the superlevel set $S_t$. Note that $|S_t| + |E_t| = |D|$. Then

$$\frac{|S_t|}{|D|} = 1 - \lambda_t \geq 1 - (t/2)^{1/k}. \tag{57}$$

□

The following theorem constructs an adversarial polynomial (i.e., has moderately large error however we compress). Remember from Appendix A that our global assumption for functions is this paper is that they have finite convergence radius, so the polynomial constructed here lies within the assumption.

**Theorem 8.** *Let $\mu$ and $\mu'$ be non-negative distributions supported on $D$: $\mu = \sum_{i=1}^d \delta_{x_i}$, $\mu' = \sum_{j=1}^{d'} c_j \delta_{y_j}$ where all $x_i, y_j \in D$ and $c_j > 0$. There exists a $\mu$ such that for any $\mu'$, there exists a polynomial $g$ and constants $A, B > 0$ such that*

$$\left|\int_D g \, d\mu - \int_D g \, d\mu'\right| \geq A d \exp[-B d'^{1/m}]. \tag{58}$$

*Proof.* Let $k$ be the smallest integer with $N_{m,k} = \binom{m+k}{m} > d'$. Let $q : \mathbb{R}^m \mapsto \mathbb{R}$ be a degree-$k$ polynomial. Since there are $N_{m,k}$ parameters in $q$, there exists a non-zero polynomial such that $q(y_1) = \cdots = q(y_{d'}) = 0$. Also, $q$ is normalized so that $\|q\|_{L^\infty(D)} = 1$. Let $g = q^2$ be the adversarial function that will be shown to satisfy Eq. (58). For $g$ and $\mu'$, we have

$$\int g \, d\mu' = \sum_{j=1}^{d'} c_j q(y_j)^2 = 0. \tag{59}$$

Next, we consider $\int_D g \, d\mu$. Denote the point set of $\{x_i\}_{i=1}^d$ by $X_d$. Quasi-uniformity of $X_d$ implies that the number of points inside a region is comparable to the volume: $\#(X_d \cap S) \geq \frac{d}{c_D}|S|$ for some constant $c_D > 0$. Let $S$ be the superlevel set $S_t$, we have

$$\int_D g \, d\mu = \sum_{i=1}^d q(x_i)^2 \geq \frac{1}{c_D} d |S_t| t^2 \geq \frac{|D|}{c_D} d\left(1 - (t/2)^{1/k}\right) t^2, \tag{60}$$

where we used Lemma 2 in the last line. We further plug in $t = 2e^{-Ck}$ to have

$$\int_D g \, d\mu \geq \frac{|D|}{c_D} d \left(1 - e^{-C}\right) e^{-2Ck}. \tag{61}$$

Putting this together with $\int_D g \, d\mu' = 0$ and denote $A = \frac{|D|}{c_D} \left(1 - e^{-C}\right)$,

$$\left| \int_D g \, d\mu - \int_D g \, d\mu' \right| \geq A d e^{-2Ck}. \tag{62}$$

Finally, recall that $k$ is the minimal integer with $N_{m,k} > d'$. Since $N_{m,k} = \binom{m+k}{m} \sim k^m/m!$, there exist constants $c_1$ and $c_2$ (depending only on $m$) such that

$$c_1 d^{1/m} \leq k \leq c_2 d^{1/m}. \tag{63}$$

Therefore,

$$\left| \int_D g \, d\mu - \int_D g \, d\mu' \right| \geq A d \exp[-B d'^{1/m}], \tag{64}$$

where $B = -2C \cdot c_2$.    $\square$

Finally, we show that Theorem 8 leads to a $\Theta((\log d)^m)$ compression lower bound. We require the compression error to be at most $\varepsilon(d)$, which is a vanishing function when $d \to \infty$. This is not possible if

$$A d \exp[-B d'^{1/m}] \geq \varepsilon(d), \tag{65}$$

which is equivalent to

$$d' \leq \frac{1}{B} \left( \log \frac{Ad}{\varepsilon(d)} \right)^m. \tag{66}$$

Usually, say $\varepsilon(d) \propto d^{-\alpha}$, the right-hand side is proportional to $(\log d)^m$. Therefore, a universal compression from $d$ objects to $O((\log d)^m)$ is optimal.

## F    FORMAL THEORY ON THE DYNAMICAL LTH

In this Appendix, we formulate all ideas mentioned in Sec. 3 with mathematical rigor.

Let $S_d$ be the permutation group of $d$ elements. Let $V = \mathbb{R}^m$. We define the representation $R : S_d \mapsto \text{End}(V^d)$ as

$$R(\sigma)(w_1, \ldots, w_d) = (w_{\sigma(1)}, \ldots, w_{\sigma(d)}). \tag{67}$$

Note that the definition of $f : V^d \mapsto V^d$ being symmetric is equivalent to: for any $\sigma \in S_d$, $f \circ R(\sigma) = f$.

**Definition 3** (Equivariant map). *A function $\mathcal{T} : V^d \mapsto V^d$ is called an equivariant map if for any $\sigma \in S_d$,*

$$\mathcal{T} \circ R(\sigma) = R(\sigma) \circ \mathcal{T}. \tag{68}$$

The dynamics induced by equivariant maps has been studied in conventional settings of dynamical systems Field (1980) but has not received much attention in deep learning. In fact, almost all update rules that we commonly use are equivariant, which we will verify for SGD and Adam later in this Appendix. Because compositions of equivariant maps are also equivariant, it follows that the entire training dynamics (i.e., the mapping from initial model parameters to trained parameters) is equivariant.

The following lemma shows that the composition of a symmetric function with an equivariant mapping is also a symmetric function.

**Lemma 3.** *If $f$ is a symmetric function and $\mathcal{T}$ is an equivariant map, then $f \circ \mathcal{T}$ is a symmetric function.*

*Proof.* For any $\sigma \in S_d$,

$$(f \circ \mathcal{T}) \circ R(\sigma) = (f \circ R(\sigma)) \circ \mathcal{T} = f \circ \mathcal{T}. \tag{69}$$

$\square$

Thanks to this lemma, we can treat $f \circ \mathcal{T}$ as a single symmetric function, and thus we expect that compressing the weights by our moment matching method induces a vanishing error in the value of $f \circ \mathcal{T}$, that is, literally any prediction of the trained model.

The following lemma establishes a general representation of equivariant maps in terms of moments, so it can be viewed as an extension of the multivariate FTSP.

**Lemma 4.** $\mathcal{T}$ *is an equivariant map if and only if for all* $i \in [d]$,

$$\mathcal{T}_i(\theta) = T(w_i, \boldsymbol{p}), \tag{70}$$

*where* $\boldsymbol{p}$ *is a collective notation for* $\{p_k = \frac{1}{d} \sum w_i^{\otimes k}\}_{k=1}^{d-1}$. *Note that* $T$ *is a function that does not depend on* $i$, *and is uniquely determined by* $\mathcal{T}$.

*Proof.* First, we verify that $\mathcal{T}_i(\theta) = T(w_i, \boldsymbol{p})$ for any function $T$ is an equivariant map. Note that $\sigma(\boldsymbol{p}) = \boldsymbol{p}$. Following the definition of equivariance,

$$\mathcal{T}_i \circ \sigma(\theta) = \mathcal{T}_i(w_{\sigma(1)}, \ldots, w_{\sigma(d)}) = T(w_{\sigma(i)}, \boldsymbol{p}) = \mathcal{T}_{\sigma(i)}(\theta). \tag{71}$$

Thus, $\mathcal{T} \circ \sigma = \sigma \circ \mathcal{T}$.

The rest of this proof is to show that all equivariant maps can be written in the asserted form. For a fixed $i$, let $\hat{\sigma} \in S_d$ be any permutation group element such that $\hat{\sigma}(i) = i$. Consider the $i$th component of the equation $\mathcal{T}(\hat{\sigma}(\theta)) = \hat{\sigma}(\mathcal{T}(\theta))$:

$$\mathcal{T}_i(\hat{\sigma}(\theta)) = \mathcal{T}_i(\theta). \tag{72}$$

This means that $\mathcal{T}_i$ is invariant under any permutation on $[d] \setminus \{i\}$, which forms a representation of $S_{d-1}$. By Theorem 3, $\mathcal{T}_i$ can be uniquely written as a function of $w_i$ and power sums $\sum_{j \neq i} w_j^{\otimes k}$. Since $\sum_{j \neq i} w_j^{\otimes k}$ is uniquely determined by $w_i$ and $p_k$ as $dp_k - w_i^{\otimes k}$, we conclude that there is a unique function $T_i$ such that

$$\mathcal{T}_i(\theta) = T_i(w_i, \boldsymbol{p}). \tag{73}$$

The final task is to show that $T_i$ does not depend on $i$. We use $\pi_{i,j}$ to denote the permutation group element that only exchanges $i$ and $j$. The $i$th component of the equation $\mathcal{T}(\pi_{i,j}(\theta)) = \pi_{i,j}(\mathcal{T}(\theta))$ reads

$$T_i(w_j, \boldsymbol{p}) = T_j(w_j, \boldsymbol{p}), \tag{74}$$

which completes the proof. □

To make the dynamical LTH practically applicable, we must define a compressed dynamics $\mathcal{T}'$, which maps compressed initial parameters to compressed trained parameters. The formal definition is given below, but in practice it is straightforward. For gradient-based update rules such as SGD or Adam, the compressed dynamics acting on $\theta' = \{(c_j, w_j)\}_{j \in [d']}$ is identical to the original dynamics on $\theta$, except that each gradient $\partial L / \partial w_j$ is replaced by $c_j^{-1} \partial L / \partial w_j$.

We use Lemma 4 to unambiguously define the compressed training dynamics:

**Definition 4** (Compressed dynamics). *Suppose a training dynamics is determined by an equivariant map* $\theta = \mathcal{T}(\theta_0)$ *(arbitrarily initialized weights to trained weights), which can be written as in Eq.* (70). *For the weighted parameters* $\theta' = \{c_j, w_j\}_{j \in [d]}$, *we define the compressed dynamics* $\mathcal{T}'$ *as*

$$\mathcal{T}'_j(\theta') = T(w_j, \boldsymbol{p}'), \tag{75}$$

*where* $\boldsymbol{p}'$ *is a collective notation for* $\{p'_k = \frac{1}{d} \sum_j c_j w_j^{\otimes k}\}_{k=1}^d$. *Note that* $\mathcal{T}'$ *is uniquely determined by* $\mathcal{T}$. *Also note that* $c_j$ *never changes with the dynamics; some* $c_j$ *might be zero so that they are actually pruned.*

The mapping from original learning dynamics to compressed ones is in fact easy in practice. Below are some common examples.

1. Stochastic gradient descent (SGD). Consider a two-layer neural network used in supervised learning. For simplicity, we write the output as $y = \sum_{i=1}^d g(w_i, x)$, which is symmetric in

$\theta = (w_1, \ldots, w_d)$. Each time we choose a batch of training data $\{(x_a, y_a)\}_{a \in \mathcal{B}}$. We denote the per-sample loss function as $\ell_a = \ell(y(\theta, x_a), y_a)$. The SGD update rule is

$$
\begin{aligned}
\mathcal{T}_i(\theta) &= w_i - \eta \mathop{\mathbb{E}}_{a \in \mathcal{B}} \frac{\partial \ell_a}{\partial w_i} \\
&= w_i - \eta \mathop{\mathbb{E}}_{a \in \mathcal{B}} \frac{\partial \ell_a}{\partial y} \frac{\partial g(w_i, x_a)}{\partial w_i}
\end{aligned}
\tag{76}
$$

where $\eta$ is the learning rate. Note that $\partial \ell_a / \partial y$ is a function of $y$, which is thus permutation invariant in $\theta$. We can explicitly compute $(\mathcal{T} \circ R(\sigma))(\theta)$ and $(R(\sigma) \circ \mathcal{T})(\theta)$, which are both equal to

$$
w_{\sigma(i)} - \eta \mathop{\mathbb{E}}_{a \in \mathcal{B}} \frac{\partial \ell_a}{\partial y} \frac{\partial g(w_{\sigma(i)}, x_a)}{\partial w_{\sigma(i)}},
\tag{77}
$$

so SGD is indeed equivariant.

Then, we derive the compressed SGD. Remember that the weighted neurons compute the output as $y = \sum_{j=1}^{d'} c_j g(w_i, x)$. Eq. (76) can indeed be written in the form $T(w_i, \boldsymbol{p})$ because $\partial \ell_a / \partial y$ is a function of $\boldsymbol{p}$, and $\partial g(w_i, x_a) / \partial w_i$ solely depends on $w_i$. Therefore, the compressed update rule still looks like

$$
\mathcal{T}'_j(\theta) = w_j - \eta \mathop{\mathbb{E}}_{a \in \mathcal{B}} \frac{\partial l_a}{\partial y} \frac{\partial g(w_i, x_a)}{\partial w_j}.
\tag{78}
$$

However, we emphasize that it is not $w_j - \eta \mathbb{E}_{a \in \mathcal{B}} \partial \ell_a / \partial w_j$, because

$$
\frac{\partial \ell_a}{\partial w_j} = \frac{\partial \ell_a}{\partial y} c_j \frac{\partial g(w_j, x_a)}{\partial w_j}
\tag{79}
$$

Effectively, whenever there is a gradient $\partial L / \partial w_j$ in the original update, we should replace it by $c_j^{-1} \partial L / \partial w_j$. This rule applies for all other gradient-based updates as well.

Finally, we remark on non-deterministic updates. It seems that choosing mini-batches turns the update into a stochastic process, which complicates the problem. But in fact, for any fixed trajectory of mini-batch choices, the update is explicitly equivariant (choosing a mini-batch breaks the permutation symmetry among the training dataset, but not the permutation symmetry of neurons). Therefore, we always expect to see good agreement between the original and compressed dynamics if we use the same choice of mini-batches for both.

2. Adam. The Adam update rule for standard (i.e., uniformly weighted) parameters reads

$$
\begin{aligned}
g_t &\leftarrow \nabla_\theta \mathop{\mathbb{E}}_{a \in \mathcal{B}} \frac{\partial \ell_a}{\partial \theta}(\theta_{t-1}) \\
m_t &\leftarrow \beta_1 m_{t-1} + (1 - \beta_1) g_t \\
v_t &\leftarrow \beta_2 v_{t-1} + (1 - \beta_2) g_t^2 \\
\hat{m}_t &\leftarrow \frac{m_t}{1 - \beta_1^t} \\
\hat{v}_t &\leftarrow \frac{v_t}{1 - \beta_2^t} \\
\theta_t &\leftarrow \theta_{t-1} - \eta \frac{\hat{m}_t}{\sqrt{\hat{v}_t} + \epsilon},
\end{aligned}
\tag{80}
$$

where $t$ denotes time step. To check that Adam is equivariant, we only need to note that the gradient $(g_t)$ for $w_i$ takes the form

$$
\mathop{\mathbb{E}}_{a \in \mathcal{B}} \frac{\partial \ell_a}{\partial y} \frac{\partial g(w_i, x_a)}{\partial w_i},
\tag{81}
$$

where $\partial \ell_a / \partial y$ is symmetric, and $\partial g(w_i, x_a) / \partial w_i$ is solely a function of $w_i$. Using this fact, it is straightforward to check that $(\mathcal{T} \circ R(\sigma))(\theta)$ and $(R(\sigma) \circ \mathcal{T})(\theta)$ are identical.

To define the compressed version of Adam, we need to keep the gradient exactly as in Eq. (81), as in our discussion on SGD. This in turn tells us that we just need to replace

$\partial L / \partial w_j$ by $c_j^{-1} \partial L / \partial w_j$. Special to Adam, because $1/c_j$ appears in both $\hat{m}_t$ and $\sqrt{\hat{v}_t}$, the compressed update rule is basically the same as the original if we neglect the small $\epsilon$. Indeed, in the numerical simulations we conducted with AdamW (the reasoning is the same as Adam), we did not observe any visible difference whether or not to scale the gradients by $1/c_j$.

Finally, we prove the dynamical LTH.

**Theorem 9** (Dynamical lottery ticket hypothesis). *Let $\theta = \{w_i\}_{i \in [d]}$ be a set of permutation-symmetric trainable parameters of a neural network, and each $\|w_i\| \leq R$. Suppose the model prediction $f : V^d \to \mathbb{R}$ is permutation invariant, and the training dynamics $\mathcal{T} : V^d \to V^d$ is equivariant. Also, suppose $f \circ \mathcal{T}$ satisfies Assumption 1. Then, for any initial parameter $\theta_0$, there exists a compressed weighted parameter $\theta_0'$ (which does not depend on $f$ or $\mathcal{T}$), supported on $d' = O(\log^m \frac{d}{\varepsilon(d)})$ points, such that*

$$\left| f(\mathcal{T}'(\theta_0')) - f(\mathcal{T}(\theta_0)) \right| = \varepsilon(d). \tag{82}$$

*Proof.* We compress $\theta_0$ using moment matching. By construction, for all $l = 0, 1, \ldots, k$ we have

$$\sum_{i=1}^d (w_0)_i^{\otimes l} = \sum_{j \in S} c_j (w_0)_j^{\otimes l} \tag{83}$$

For any $f$, $f \circ \mathcal{T}$ and $f \circ \mathcal{T}'$ can both be written as a function of moments. In this representation, using the definition in Eq. (75), one can check that they are exactly the same function. The difference is that $f \circ \mathcal{T}$ takes in $(p_0)_k = \frac{1}{d} \sum_i (w_0)_i^{\otimes k}$, while $f \circ \mathcal{T}'$ takes in $(p_0')_k = \frac{1}{d} \sum_j c_j (w_0)_j^{\otimes k}$; they are identical in the first $k$ moments by construction. Using Theorem 6, we conclude that using large enough $k$, the difference between $f(\mathcal{T}(\theta_0))$ and $f(\mathcal{T}'(\theta_0'))$ can always be made vanishing, with the same error upper-bound as asserted in Theorem 6. Ultimately, using the optimal $k_{\text{opt}}$ given in Theorem 7, we achieve $d' = O\left(\log^m \frac{d}{\varepsilon(d)}\right)$ with error at most $\varepsilon(d)$. $\qquad \square$

## G   DETAILS ON NUMERICAL EXPERIMENTS

We studied two main numerical tasks in Figs. 2, 3 and 4. For all these experiments, the loss function in training is the mean squared error (MSE) loss. The test loss shown in figures is the MSE loss evaluated on a randomly sampled dataset of the form $(x_1, x_2, f(x_1, x_2))$, where $f(x_1, x_2)$ is the ground truth function. For Figs. 2 and 3, the test dataset size is $10^5$, and for Fig. 4 it is $2 \times 10^4$. All unspecified training hyperparameters follow PyTorch defaults. In particular, for AdamW, they are $\beta_1 = 0.9$, $\beta_2 = 0.999$, (weight decay) $\lambda = 0.01$, and $\epsilon = 10^{-8}$.

Figures 2 and 4(a) concern compressing the training dataset. We study a supervised learning problem in a teacher–student setup. Specifically, the teacher $f(x_1, x_2)$ is implemented as a random two-layer neural network of width $50$ with ReLU activation. The student receives $d$ data points of the form $(x_1, x_2, \mathcal{N}(f(x_1, x_2), 3^2))$, where $x_1$ and $x_2$ are uniformly sampled from $[-1, 1]$, and the goal is to learn $f(x_1, x_2)$ using an identical network architecture. When the training dataset is weighted, the data loader is implemented as follows: We draw i.i.d. samples from $\{w_j\}_{j \in [d']}$, using $\{c_j\}_{j \in [d']}$ as the unnormalized probability distribution, to form a mini-batch.

The loss function is exactly symmetric in the dataset. For full-batch update rules, which depend on the gradient of the full-batch loss, any model prediction or performance metric is thus a symmetric function of the dataset. Consequently, if the student is given a compressed dataset of size $d'$ derived from an original dataset of size $d$, the performance approximates that of training on the full dataset, and typically surpasses that of training on $d'$ naively subsampled data points (Fig. 2(a,b)).

In practice, however, stochastic update rules based on mini-batches are more common. In this case, permutation symmetry holds only in an averaged sense. Nonetheless, we find that compression remains useful. Figures 2(c,d) show the loss evolution with batch size 256, supporting this claim.

Figures 3 and 4(b) concern compressing the width of a two-layer neural network. The task is fitting an oscillating bivariate function known as a cylindrical harmonic, described in the caption of Fig. 3. The training dataset size for Fig. 3 is $10^5$, and for Fig. 4 is $2 \times 10^4$.

In Fig. 4, we show the test MSE loss scaling with respect to training dataset size (a) and neural network width (b). For both (a) and (b), the update rule is AdamW. The learning rate is initially $0.001$, and is modulated by a cosine annealing learning rate scheduler, reaching $0$ at the final epoch. Each data point is obtained by averaging 10 random instances of the train and test dataset and the neural network initialization, and the error bars indicate the standard deviation. For (a), we train each instance for 2048 epochs, each epoch containing one mini-batch of size 512, so that there is a constant compute budget for the original and the compressed datasets. For (b), we train each instance for 2000 epochs, each epoch enumerates over the train dataset. The batch size is 128.

## H  PERMUTATION SYMMETRY IN ATTENTION MODULES

In principle, the compression theory developed in this work can be applied to attention mechanisms in two distinct ways. We briefly alluded to these ideas in Section 2; here, we provide a self-contained and more detailed discussion. The two applications concern $(1)$ the compression of the query and key weight matrices, and $(2)$ the compression of attention heads.

**Compression of query and key matrices.**    The first application serves as a straightforward verification of the theory. Consider the query and key matrices $W_Q$ and $W_K$. The attention logits depend only on their product, which can be written as

$$a = a(W_Q W_K) \tag{84}$$

with

$$W_Q W_K = \sum_{i=1}^{d} w_Q^i \left( w_K^i \right)^T, \tag{85}$$

where $w_Q^i$ denotes the $i$-th row of $W_Q$ and $w_K^i$ the $i$-th column of $W_K$. Since the index $i$ is a dummy summation index, its ordering is immaterial. This reveals an explicit permutation symmetry among the pairs $\{(w_Q^i, w_K^i)\}_{i=1}^{d}$.

Because the output depends only on the sum of these outer products, the symmetry implies that the collection of these row–column pairs can be compressed. Moreover, if the left dimension of $W_Q$ is $m$, then $W_Q W_K$ has rank at most $m$, and the effective number of degrees of freedom is independent of $d$. Consequently, one can achieve not merely a $\mathrm{polylog}(d)$ compression but, in fact, a constant-size representation. This serves as a useful sanity check for the consistency of our general theory.

A more interesting direction arises when the key–query interaction becomes nonlinear. For instance, one may consider replacing the bilinear form with

$$\sum_{i=1}^{d} w_Q^i s\left( w_K^i, X \right)^T, \tag{86}$$

where $s$ is a nonlinear function and $X$ denotes the input data. In this setting, the permutation symmetry persists, and our theory guarantees that an $\mathrm{polylog}(d)$-size compressed representation of this nonlinear attention layer is achievable in principle.

**Compression of attention heads.**    The second application concerns the compression of entire attention heads, which is intrinsically more meaningful. Consider an attention layer with $d$ heads, and let

$$A_i = B(w_i, X) \tag{87}$$

denote the output of the $i$th head, where $w_i$ denotes its trainable parameter (including the query/key/value weight matrices $W_{Q/K/V,i}$ in the $i$th head) and $B$ the head-level transformation. The standard output (denoted by $y$, of the attention layer is

$$y = U \operatorname{concat}(A_1, \ldots, A_d), \tag{88}$$

where $U \in \mathbb{R}^{z \times dh}$ is the output projection matrix, $z$ is the dimension of the final output, and $h$ is the dimension of each head output. Partitioning $U$ into blocks $U_i \in \mathbb{R}^{z \times h}$, this expression becomes

$$y = \sum_{i=1}^{d} U_i B(w_i, X). \tag{89}$$

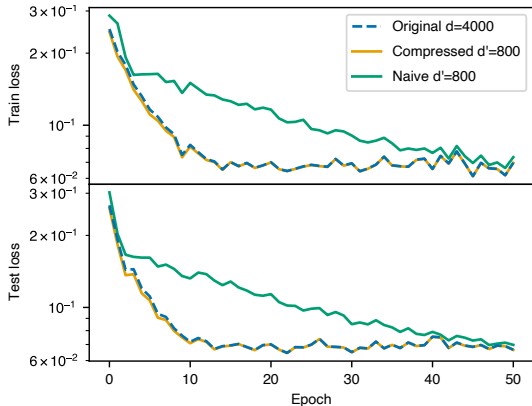

Figure 6: Dynamical LTH extended to transformers. The training dynamics of a large, $d_{\text{heads}}$ = 4000-head attention model shows good agreement with its compressed multi-head attention model with $d'_{\text{heads}}$ = 800 heads. See description of the task in Appendix H.

This summation structure again exposes a permutation symmetry: the parameters

$$\theta_i = (U_i, w_i) \tag{90}$$

enter the model only through their sum over $i$, and their ordering is irrelevant. By the general theory, any such collection of $d$ symmetric objects admits a compressed representation of size $O(\text{polylog}(d))$. Hence, the entire set of attention heads can be compressed to $\text{polylog}(d)$ effective parameters while preserving the functional form of the output.

With the permutation symmetry among heads, it is easy to formulate a similar dynamical LTH for multi-head attention. Figure 6 is a numerical demonstration of LTH in transformers. Here, the task is in-context learning on random piecewise-linear functions. We consider a scalar in-context regression task in which each episode defines a random continuous piecewise-linear function $f : [0, 1] \to \mathbb{R}$. For a given episode, we first draw an initial value $f_0 \sim \mathcal{N}(0, \sigma_{f_0}^2)$ and segment slopes $s_j \sim \mathcal{N}(0, \sigma_s^2)$ for $j = 0, \ldots, K - 1$, with $K = 16$, $\sigma_{f_0} = 0.5$ and $\sigma_s = 1.0$. The interval $[0, 1]$ is partitioned into $K$ equal sub-intervals of length $1/K$, and $f$ is defined by enforcing continuity and setting the slope on segment $j$ to $s_j$. For each episode we sample $n_{\text{ctx}} = 8$ context inputs $x_i \sim \text{Unif}[0, 1]$ with noisy observations $y_i = f(x_i) + \varepsilon_i$, where $\varepsilon_i \sim \mathcal{N}(0, \sigma_{\text{noise}}^2)$ with $\sigma_{\text{noise}} = 0.3$, together with an additional query point $x_* \sim \text{Unif}[0, 1]$ and a clean target $y_* = f(x_*)$. The episode is presented to the model as a scalar sequence of tokens $[x_1, y_1, \ldots, x_{n_{\text{ctx}}}, y_{n_{\text{ctx}}}, x_*] \in \mathbb{R}^{2n_{\text{ctx}}+1}$ (token dimension $d_{\text{in}} = 1$), which is processed by a single-layer causal multi-head attention module with $d_{\text{heads}} = 4000$ heads and per-head dimension $d_{\text{head}} = 2$. The model outputs a scalar prediction $\hat{y}_*$ from the final token position. We compare three variants that share the same initialization: (i) the full model with all 4000 heads, (ii) a compressed model obtained by reducing the number of heads to $d'_{\text{heads}} = 800$ via compression of order $k = 3$ (the effective dimension of each symmetric object is $m = 8$), and (iii) a naive head-pruned model where 800 heads are selected uniformly at random and the remaining heads are discarded. All three models are trained with Adam (learning rate $10^{-4}$, batch size 256) on the same sequence of mini-batches, using 5 gradient steps per epoch for 50 epochs. At epoch 0 (before training) and after each epoch we report the MSE loss, for both training-like and test-like evaluations.

## I FURTHER NUMERICAL VALIDATIONS

### I.1 ERROR SCALING WITH DIFFERENT DATA DIMENSIONS AND MATCHING ORDER

Figure 7 presents a direct numerical test of the compression error scaling predicted in Theorem 6. We study the following function:

$$f(w_1, \ldots, w_d) = \frac{1}{d} \sum_{i=1}^{d} \frac{1}{10} \sum_{a=1}^{10} \text{sigmoid}(\langle w_i, x_a \rangle), \tag{91}$$

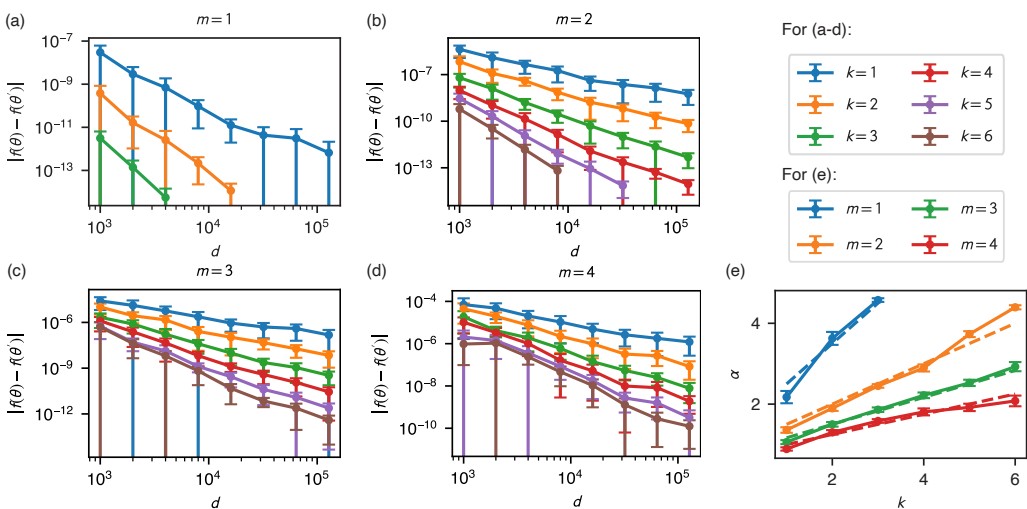

Figure 7: Error scaling for compressing a general symmetric function (Eq. (91)) using the moment-matching method. (a–d): each point shows the error in $f$ after compressing $d \to \max([0.1d], N_{m,k})$ input objects. Matching higher-order moments leads to faster error decay. (e): $\alpha$ is the fitted exponent in $|f(\theta) - f(\theta')| \propto d^{-\alpha}$. The dashed lines indicate $(k+1)/m + 0.5$, which show good agreement with the numerical results.

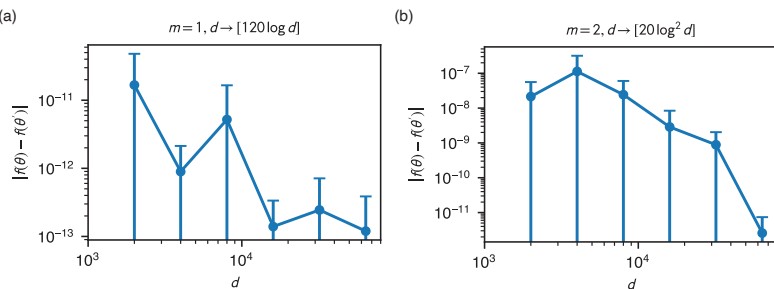

Figure 8: Error scaling of compressing a general symmetric function using the moment-matching method. Here, various different values of $k$ (the order of moment matching) are attempted, from small to large, until the smallest error is found. Each data point is an average over 5 random instances, plotting the average with error bar standing for one standard deviation.

where all $w_i$ and $x_a$ are $m$-dimensional vectors, and $\langle \cdot, \cdot \rangle$ denotes the inner product. We compress a fixed fraction of the original dataset across different dimensions and with varying moment-matching orders. Specifically, we randomly sample vectors $x_a$ and inputs $\{w_i\}_{i \in [d]}$, perform compression, and average results over 20 trials to obtain each data point in Fig. 7 (a–d); error bars indicate standard deviation. Figure 7(e) shows that the error decays approximately as $\mathcal{E} \sim d^{-(k+1)/m-0.5}$. While Eq. (30) predicts an upper bound $\mathcal{E} \sim d^{-(k+1)/m+2}$, the observed error lies well below this bound and shows a dependency on $k$ and $m$ that is similar to the theoretical bound.

## I.2   COMPRESSION TO $\mathrm{polylog}(d)$

In this section, we numerically demonstrate the possibility of compressing to the optimal rate, that is, compressing $d$ objects into $O(\log^m d)$ objects. As we argued in Theorem 7 and Appendix D.1, compressing to this rate is computationally heavy for the moment-matching algorithm, so we only show it in low dimensions, i.e., $m = 1$ or 2.

The error scaling is shown in Fig. 8. The function we study is

$$f(w_1, \ldots, w_d) = \frac{1}{d} \sum_{i=1}^{d} \frac{1}{10} \sum_{a=1}^{10} \frac{1}{A + \langle w_i, x_a \rangle}. \tag{92}$$

For $m = 1$ (Fig. 8(a)), we use $A = 1.05$; for $m = 2$ (Fig. 8(b)), we use $A = 2.05$. In Fig. 8(a), we plot the error of compressing $d$ 1d random objects into $\lceil 120 \log d \rceil$; in Fig. 8(b), we plot the error of compressing $d$ 2d objects into $\lceil 20 \log^2 d \rceil$. All unspecified setting of this numerical experiment is identical with that of Fig. 7. Despite pruning bigger fraction of the objects when $d$ increases, we see that the error is not increasing with $d$, but rather overall vanishing with $d$. Thus, this shows that it is possible to compress $d$ to $O(\log^m d)$ objects losslessly. However, the numerical error shows visible oscillation, possibly due to the volatile moment-matching order $k_{\text{opt}}$ and finite-size ($d$) effect.

