# OpenReview forum: "A universal compression theory for lottery ticket hypothesis and neural scaling laws"
_ICLR.cc/2026/Workshop/GRaM — ICLR 2026 Workshop GRaM Poster_

### Official Review · Reviewer_rHTM · 2026-02-22
**Universal compression via permutation symmetry provides a promising theory with practical connections to scaling laws**

**Rating:** 8
**Confidence:** 2

**Review:**

This work proposes a “universal compression” theory for permutation-invariant functions of $d$ objects. The core claim is that, for a broad function class, one can replace those objects by a weighted subset of size $\text{polylog}(d)$ while preserving all such symmetric function evaluations up to vanishing error. The authors argue that this implies a dynamical variant of the lottery ticket hypothesis, and an interpretation of how compression could accelerate observed scaling laws when measuring performance vs. effective size.

**Strengths**.
* Strong theoretical result: The authors present a clean theoretical analogue of a dynamical lottery ticket hypothesis under permutation symmetry and analyticity assumptions.
* Unifying viewpoint: Treating both dataset elements and “exchangeable” parameter groups as objects under permutation symmetry provides a clean conceptual bridge between data compression and model compression.
* Initial empirical sanity checks: The work also draws out concrete practical implications that follow naturally from the theoretical results rather than being loosely asserted. The plots compare compressed vs. original vs. “same-size random subset/subnetwork,” which is a reasonable baseline for showing the compression is doing something nontrivial in these toy regimes.

**Weaknesses**.
* The compression size scales like $ (\log d)^m$, where m is the object dimension. For datasets (images, text embeddings) and many parameter blocks, $m$ can be very large. This could make the worst-case guarantee potentially impractical.

**Pmlr Suitability:**

NA

---

### Official Review · Reviewer_8par · 2026-02-24
**Dynamic lottery ticket hypothesis and data compression from permutation symmetries**

**Rating:** 8
**Confidence:** 4

**Review:**

This paper develops a theoretical framework for compressing both neural networks and datasets by exploiting permutation symmetry. The main claim is a universal compression theorem: a generic permutation-invariant function of $d$ "objects" ( datapoints, neurons/weights grouped into exchangeable units) can be asymptotically represented using only $\operatorname{polylog}(d)$ objects with vanishing error.

The authors formalize  learning objectives as symmetric functions of $d$ exchangeable objects, and relate compressibility to the fact that symmetric functions can be characterized through low-order moments of the underlying objects. They propose a multivariate variant of the fundamental theorem of symmetric polynomials that connects symmetric polynomials to moment tensors and argue that matching finitely many moments can be achieved with a much smaller weighted support.

The authors draw two main implications:

- The author propose a **dynamical** variant of the **lottery ticket hypothesis** (LTH), that a dense network can be compressed to polylogarithmic width while **preserving training dynamics** (not just final performance), i.e., learning trajectories before and after compression are identical (under the framework’s conditions).
-  A dataset can be compressed to polylogarithmic size while leaving the resulting loss landscape unchanged, which the authors argue boosts neural scaling laws from power-law decay $L\sim d^{-\alpha}$ toward much faster  behavior in the idealized setting.



This contribution is well written and proposes a bold and interesting claim that permutation symmetry alonge can provide means for such a dramatic scale compression with implications about training dynamics and data compression.
However, the strongest claims hinge on how the theorem’s assumptions and the exact notion of training dynamics preservation translate to practical deep learning settings.
I think this is a valuable contribution for the workshop.

**Pmlr Suitability:**

NA

---

### Official Review · Reviewer_ZM8W · 2026-02-28
**Ambitious Symmetry-Based Compression Theory with Limited Large-Scale Validation**

**Rating:** 7
**Confidence:** 4

**Review:**

The paper proposes a universal compression theorem showing that permutation-invariant functions over d objects can be approximated using only polylog(d) weighted objects with vanishing error, with applications to neural network compression, a dynamical lottery ticket hypothesis, and improved neural scaling laws.

The paper is ambitious and theoretically interesting. The main contribution is a symmetry-based compression framework grounded in moment matching and Tchakaloff-type results, with clear implications for model and dataset compression. The connection to a dynamical version of the lottery ticket hypothesis is particularly novel, and the theoretical development is mathematically detailed and self-contained.

However, several limitations remain. First, many results are stated informally in the main text, with key assumptions (e.g., analyticity, finite radius of convergence, bounded norms) playing a crucial role but not fully discussed in terms of practical relevance. Second, while the theory suggests dramatic improvements to neural scaling laws, the empirical validation is limited to relatively small synthetic or teacher–student setups, leaving open whether such compression is feasible or stable in large modern architectures.

Overall, the work presents an intriguing and mathematically rich perspective on symmetry and compression, but the gap between theory and large-scale practical impact weakens the strength of its claims.

**Pmlr Suitability:**

Yes

---

### Meta-Review · Area_Chair_yuA5 · 2026-02-26

**Decision:**

Accept

**Metareview:**

This paper proves that permutation symmetry enables polylogarithmic compression of neural networks and datasets, thus establishing the dynamical lottery ticket hypothesis and boosting neural scaling laws. Reviewers find the theoretical results strong and appreciate the unifying viewpoint and concrete practical implications. I recommend acceptance and encourage the authors to incorporate reviewers’ feedbacks in the final version.

**Relevance To Proceedings:**

Tiny paper — does not apply

**Relevance To Workshop:**

Yes — suitable for GRaM

---

### Decision · Program_Chairs · 2026-03-02

Accept (Poster)